# Emerging Genetic Tools to Investigate Molecular Pathways Related to Heat Stress in Chickens: A Review

**DOI:** 10.3390/ani11010046

**Published:** 2020-12-29

**Authors:** Francesco Perini, Filippo Cendron, Giacomo Rovelli, Cesare Castellini, Martino Cassandro, Emiliano Lasagna

**Affiliations:** 1Department of Agricultural, Food and Environmental Sciences, University of Perugia, Borgo XX Giugno, 74, 06121 Perugia (PG), Italy; francesco.perini@studenti.unipg.it (F.P.); giacomo.rovelli@studenti.unipg.it (G.R.); cesare.castellini@unipg.it (C.C.); emiliano.lasagna@unipg.it (E.L.); 2Department of Agronomy, Food, Natural Resources, Animals and Environment, University of Padova, Viale dell’Università, 16, 35020 Legnaro (PD), Italy; martino.cassandro@unipd.it

**Keywords:** biodiversity, poultry, heat shock protein, climate change

## Abstract

**Simple Summary:**

New genomic tools have been used as an instrument in order to assess the molecular pathway involved in heat stress resistance. Local chicken breeds have a better attitude to face heat stress. This review aims to summarize studies linked to chickens, heat stress, and heat shock protein.

**Abstract:**

Chicken products are the most consumed animal-sourced foods at a global level across greatly diverse cultures, traditions, and religions. The consumption of chicken meat has increased rapidly in the past few decades and chicken meat is the main animal protein source in developing countries. Heat stress is one of the environmental factors which decreases the productive performance of poultry and meat quality. Heat stress produces the over-expression of heat shock factors and heat shock proteins in chicken tissues. Heat shock proteins regulate several molecular pathways in cells in response to stress conditions, changing the homeostasis of cells and tissues. These changes can affect the physiology of the tissue and hence the production ability of chickens. Indeed, commercial chicken strains can reach a high production level, but their body metabolism, being comparatively accelerated, has poor thermoregulation. In contrast, native backyard chickens are more adapted to the environments in which they live, with a robustness that allows them to survive and reproduce constantly. In the past few years, new molecular tools have been developed, such as RNA-Seq, Single Nucleotide Polymorphisms (SNPs), and bioinformatics approaches such as Genome-Wide Association Study (GWAS). Based on these genetic tools, many studies have detected the main pathways involved in cellular response mechanisms. In this context, it is necessary to clarify all the genetic and molecular mechanisms involved in heat stress response. Hence, this paper aims to review the ability of the new generation of genetic tools to clarify the molecular pathways associated with heat stress in chickens, offering new perspectives for the use of these findings in the animal breeding field.

## 1. Introduction

Poultry, especially chicken, is one of the most reared species in the world. Indeed, chickens have a pivotal economic and ecological role in the agriculture system and, additionally, are used in the households of developed countries as the main dietary source of animal protein. The consumption of chicken products has increased rapidly in the last few decades. Moreover, the most globally consumed products of animal origin are poultry products, as they do not face any religious or cultural restrictions [1]. Indeed, poultry is the fastest growing animal product, especially in developing countries. Chicken products have conquered the meat market, as they are commonly affordable and low in fat [2,3]. Data from 2018 showed that chicken products had dominated the market in both the egg branch and the meat industry, with 114 million tons being produced [4]. For these reasons, it is appropriate to assume that there will be an expansion of the whole poultry branch, as demand for meat and eggs is led by rising human population and urbanization. In these circumstances, the sector has to withstand unparalleled challenges [5]. In this scenario of expected growing demand [6], livestock production is likely to be undermined by climate change, competition for land and water, and food security issues [7,8]. In particular, climate change is involved in the reduction in pasture and water availability, the onset of new diseases, increasing numbers of drought seasons, and the occurrence of exceptional natural phenomena, which all reduce the quantity and quality of meat and eggs [9,10,11,12]. The alteration of environmental factors such as sunlight, temperature, and humidity affect animal metabolism and mechanisms of thermo-regulation and can cause imbalances in animal physiology [13]. Animals have proved to be adaptable in order to overcome adverse climatic conditions [14], and this permits them to survive in harsh climatic conditions [15]. Several mechanisms are the basis of this adaption ability, namely, morphological, anatomical, behavioral, physiological, biochemical, cellular, and molecular characteristics [16]. Broilers have reached high productive levels due to genetic improvements; however, their lack of effective thermoregulation and heightened metabolism entail a worse adaptation to harsh environments [17]. In contrast, native chickens generally have genetic and morphological features and rusticity that help them to face harsh conditions. Native chickens have a greater capacity to cope with heat stress [18]; however, they give a lower rate of production, since they have never been subject to breeding programs.

The aim of the present review is to summarize papers concerning the identification of heat stress (HS) genetic markers in chickens using new molecular tools. In fact, both genotypic and phenotypic traits may be useful indicators of heat stress (HS) susceptibility or tolerance in different chicken breeds. Nowadays, the identification and the inclusion of appropriate biomarkers in breeding programs for HS responses in chickens could be a crucial aspect in developing thermo-tolerant breeds using marker-assisted selection [19].

## 2. Climate Change and HS in Chickens: An Overview

Climate change means spatial and temporal variations in the environmental climatic parameters on Earth [20]. These climatic parameters include temperature; humidity; solar radiation; precipitation; and the temperature of water, which causes glaciers to melt [21]. The Intergovernmental Panel on Climate Change’s Fifth Assessment Report shed light on a future temperature increase in the Earth’s surface ranging from 0.3 to 4.8 °C by 2100 [22]; this may affect livestock through variations in feed crops and forage, biodiversity, animal welfare, productive, and reproductive traits.

Figure 1 shows climate change in the last century on the global temperature level.

The change that is occurring in the climate is mainly related to the constant increase in the level of greenhouse gasses (GHGs) [23]. All over the world, climate change is driving temperature rises, altered photoperiods, and reductions in precipitation, causing reduced feed quality and quantity [24]. Accordingly, climate change is one issue that the livestock sector will have to cope with in the following years. Above all, HS is the worst environmental stressor for poultry production worldwide. As early as 2003, a study estimated the annual economic loss due to HS in the US livestock industry to range from $1.69 to $2.36 billion; from these data, $128 to $165 million was related to the poultry sector [25,26]. Moreover, as Figure 2 shows, chicken production is mainly distributed in tropical areas, and hence, is directly affected by hot temperatures.

Exposure to HS can decrease growth rate and feed efficiency, alter immune response, damage gut microflora, and finally decrease meat quality [27]. Chickens are homoeothermic animals, able to maintain a relatively constant inner temperature, even if only in an environmental temperature of 21–28 °C. Body metabolism is the main aspect responsible for heat production, and needs heat removal to maintain the internal body temperature at 41.4–42.9 °C [28]. Indeed, hot temperatures can ruin the maintenance of this homoeothermic due to the physiological changes that affect body temperature stability. The deterioration of the regulatory mechanism of homoeothermic balance has direct and indirect consequences on chicken production. As reported in the review of Zaboli et al. [27], the negative effect of HS on meat quality is due to the combined effects of anaerobic glycolysis acceleration (metabolic acidosis) by panting [29], the increase in Reactive Oxygen Species (ROS) [30], and the corticosterone concentration in the blood [31,32].

HS occurs when the amount of heat produced by the animal surpasses its capacity to dissipate the extra heat to the surrounding environment [33]. Heat swap between the animal body and environment represents the major system to preserve homoeothermic balance. Heat generation, maintenance, and dissipation are ruled by biological and physical components. The intricacy of the phenomena requires a thermal index that could represent the environmental pressure on the biological aspects involved. The most common index by which to measure the impact of environment on poultry livestock is given by the Temperature Humidity Index (THI) [34]. The THI is a useful instrument with which to measure livestock productivity response as a function of climate [35]. THI takes into consideration the air temperature and relative humidity of the environment, and how they impact on diverse species [36], and it is an indicative measure of the sum of forces external to the animal that act to displace body temperature from its set point [37].

During the period of heat stress, chickens can employ some physiological heat exchange mechanisms in order to overcome the stressed status. First of all, birds change their behavior in HS conditions: less time is devoted to walking and standing [38], the consumption of feed is less, and the consumption of water is greater [39]. In an anatomical way, chickens are able to dissipate heat by radiation and conduction through an increase in blood flow to the skin surface. Moreover, partial feather loss is not uncommon in hens, mainly from the neck, back, and breast regions. Furthermore, painting is one of the visible response of poultry during exposure to heat. This specialized form of respiration dissipates heat by evaporative cooling at the surfaces of the mouth and respiratory passageways [40].

## 3. The Study of Heat Adaptation: New Perspectives through Genomic Tools

Over the past few decades, breeding practices have taken into consideration the estimated breeding values from a phenotypic and pedigree database, which has surely enhanced specialized chicken breeds in terms of production rate [41,42]. However, these approaches were limited according to the slow upgrade in chicken breeding and difficulty in splitting preferred from undesired traits [43]. To overcome these issues, Quantitative Traits Loci (QTLs) have been mapped based on both microsatellite and Single Nucleotide Polymorphism (SNP) marker technology in order to associate favorable phenotypic traits with genomic regions. Unfortunately, the QTL accuracy was low and the typical confidence range of the QTL usually covers a broad region or even a whole chromosome (CHR) [44]. Therefore, it was improbable that QTL mapping could highlight the genomic pathways related to the target traits in chicken [45]. Many QTLs have been associated with a phenotype of interest, such as body temperature (Appendix A).

The managing of new traits, such as HS resistance, has brought about the development of new technologies to investigate genetic relations with these phenotypes. A good approach to identify the functional genes and polymorphisms correlated with HS resilience has been accomplished by virtue of high-throughput screening technologies, such as genome-wide analyses of genetic variations [46,47]. The Genome-Wide Association Study (GWAS) was developed to shed light on the genetic background of phenotypic traits. GWAS is based on the hypothesis of links between genetic variants and different phenotypic traits [48]. This approach is favored over other genomic tools for its high power in detecting polymorphisms and mutations and its capability to find exact genomic regions [49]. This method has been used for several domestic animals, and considers Single Nucleotide Polymorphisms (SNPs) and whole-genome sequencing (WGS) [44]. GWAS is a powerful approach with which to identify candidate genes, genomic regions, and polymorphisms related to specific phenotypic features. Its application has become increasingly widespread because it is useful for more accurate breeding programs for chickens and other animals [50,51].

Another high-throughput screening technology largely used is RNA sequencing (RNA-Seq). RNA molecules carry the transcribed information encoded in selected genes, which can be translated into proteins or used to check gene expression directly (i.e., miRNA) or indirectly (i.e., heterogeneous nuclear ribonucleoproteins—hnRNPs). Hence, the rate of RNAs expressed in a known circumstance shows the cell state and can disclose stress mechanisms. RNA-Seq is used in the determination of a specific trait thought the comparison of differential gene expression among different tissues [52]. Currently, RNA-Seq is the most suitable method for studying gene expression and identifying novel RNA species. Differently from the DNA microarray method (the previously used technique), RNA-Seq results in higher resolution data and a greater dynamic range of detection [53]. Moreover, RNA-Seq is a direct method with which to obtain information about sequence identity, which is important for the study of unknown genes and novel transcript isoforms. The greatest advantage of RNA-Seq, its high resolution, has promoted transcriptomic research, generating an enormous quantity of data [52].

## 4. Limitation of High-Throughput Screening Technologies

In spite of the general success of GWAS methodology, important limitations and failures are still present [54]. Since WGS is still expensive for mass application, genomic studies are mostly based on SNP arrays, in which only a small number of polymorphisms are drafted to be highly polymorphic and are used to represent the whole genome. Furthermore, it has been suggested that many polymorphisms associated to quantitative traits are rare variants and could be missed in low-density SNP arrays, but we expect to find them in WGS data [55]. While WGS has a greater depth of analysis, at the same time the expensive cost and the need to manage a huge amount of output data make WGS a methodology that is difficult to pursue and not within everyone’s reach. First, the identification of alleles linked to a specific traits or phenotypes requires huge sample sizes (100 to 1000) for the adequate statistical power of GWAS [56]. Hence, a lot of care is needed in all the steps of the experiment (i.e., sample collection, genotyping, data pruning, software analysis, and interpretation of results) for protecting against erroneous findings. Another difficulty of GWAS is the interpretation of individual associated variants. Even though many variants are assayed, these represent only a partial slice compared to the total genetic variation. A given gene might have a greater number of polymorphisms than those reported in the chip panel usually used for GWAS studies. The disadvantage at this stage is in representing the real quantity of SNPs in the exons of the genes (hence, the coding part of the genes) [57]. In contrast, genetic variants fall within the introns, which could affect gene expression, alternative splicing, and DNA methylation, which represent a large portion of the associated loci and may be located at various physical distances from their targets. In this context, it is really hard to evaluate how genetic variants can affect the related phenotypic trait [58].

Some limitations are also highlighted in RNA-Seq technology: when an RNA-Seq assay is running, the greater part of the transcript (around 70%) may be related to a very small number of genes [59]. Although investigating gene expression levels is relatively easy, the analysis of allelic imbalance or splice variation is more difficult. This trouble is caused by using a fractional dataset from a given gene, coming from the SNPs already known or the exon/exon junction of interest, which enhances the risk of the poor statistical power of the method. To overcome these issues, an increase in the depth of reads is required, which can bring to light the transcriptome of poorly expressed genes. Unfortunately, these methods are hard to undertake in economic terms, but there are several ways to alleviate the problem. A stratagem may be to precipitate the not-target transcripts using probes when the assay has been set on specific transcripts [60], in order to ensure that the RNA-Seq is mostly performed on targets. One of the other issues is the read length. In fact, when the reads are shorter than the target, alignment and reassembly are required. In sequencing the whole transcript in a long read, the accurate sequence would be pinpointed; in this case, the analysis of allele-specific expression will be more accurate and mutually exclusive exons in transcripts will be highlighted [61]. Lastly, the issue of RNA-Seq is the validation of the experiment; indeed, it is largely known that genes characterized as divergent expressed by an RNA-Seq experiment need to be validated by real-time PCR. This additional step increases the time and cost of the assay [62].

## 5. Phenotypes against HS

Poultry, like other animals, has the possibility to develop specific phenotypes advantageous for adaption to the harsh environment where they live. There are several phenotypes that mainly act for the alleviation of HS, mostly related to feather types [63]. Indeed, feathers guarantee a thermal shelter between the animal body and the environment. Plumage delays the process of heat elimination from the skin surface [64]. As an example, the Naked Neck (Na) chicken shows a better fitness under HS conditions [65]. The Na gene is a single dominant autosomal gene that allows decreasing feathers in the neck region, which permits a better dissipation of heat [66]. The Na gene reduces the plumage cover by 20% and 40% in heterozygous and homozygous Na, respectively [67]. In broilers, it is correlated with an increase in breast muscle and body weight [68,69], a reduction in abdominal fat [70], and a reduction in body temperature [71]. Moreover, laying Na chickens under hot temperatures showed an improvement in egg mass, deposition rate, and egg characteristics [72].

In a recent study, Galal et al. [73] compared the expression levels of *HSP70* (used to assess the heat tolerance) in three Egyptian local breeds (Fayoumi, Dandarawi, and Sinai) with and without the Na gene and under normal and HS conditions. As expected, they found higher *HSP70* expression levels in crossbreeds with the Na gene, suggesting that the Na gene is responsible for the up-regulation of *HSP70* expression and has a positive impact on HS adaptation not only by reducing feather cover [73].

Another important feature of the plumage is the color of feathers, which can impact the ability of chickens to respond under HS conditions. A recent study demonstrated that dark chicks showed a lower expression rate of genes belonging to pathways of stress (cellular stress: *SOD2* and *HSPA8*; DNA damage repair: *ALKBH3*) than paler chicks [74]. This happens because the plumage element reduced solar heat gain by 5% in both light and dark plumages. However, because the overall levels of solar heat gain were greater in dark versus light plumages, there were different fractional decreases in the heat load for light (41%) versus dark (25%) plumages [64].

Lastly, Jiang et al. highlighted the importance of density in regard to plumage. Indeed, the density of contour feathers was significantly correlated to heat tolerance under acute HS, indicating that it could be regarded as a phenotypic marker for heat tolerance in chickens [75].

## 6. Physiological Biomarkers for HS Detection

Recording animal performance under HS is a method to measure an animal’s capacity to face heat. The most recorded parameters are body temperature, respiration rate, and the blood level of cortisol, which could be useful as discriminant criteria in order to identify animals that are tolerant to HS conditions [76,77]. About 50 years ago, body temperature was suggested as the most valuable HS indicator [78]. Body temperature is assessed through Rectal Temperature (ReT). The range of physiological ReT in chickens is between 40.6 and 43 °C. In 2013, Chen et al. established a precise method to evaluate HS resilience through ReT. The correlation between ReT and HS survival time was closely related at 18 h of heat exposure. Therefore, in the choice of heat resistance, it is recommended to evaluate the ReT after 18 h HS and calculate the ΔT18 (interval between the ReT of 18 h and the initial ReT) and standard deviation (SD). Animals with less than the mean value −1/2 SD could be selected as heat tolerant [79]. Unfortunately, Van Goor et al. reported very low hereditability values for many QTLs associated with body temperature [80].

HS in chickens is regulated by a huge number of organized reactions that are difficult to measure in a few in vivo biochemical or biophysical indices over a short time period. Moreover, it has been demonstrated that chicken body weight and genotype could affect the ReT [81,82]. Nowadays, one of the common methods used to identify ongoing stress status in chickens is the measurement of the corticosterone level in the blood [83,84,85,86]. Corticosterone affects stress-induced responsiveness and can protect or destroy the coping ability of the organism [87]. High levels of corticosterone occur in chickens during HS as a defense mechanism [88]. Currently, corticosterone represents the “gold standard” of stress markers in poultry, but its reliability and validity have recently been questioned. Indeed, the assay for its identification can give a false positive due to the circadian rhythm and cross-reactivity with other glucocorticoids [89].

## 7. Modulation of Inflammation and Immunity during HS

In poultry, the primary production and differentiation of immune cells and antibodies occurs in immune organs such as thymus, the bursa of Fabricius, and the spleen. HS causes oxidative damage to the cell membranes in immune organs [90]; many studies have reported significant reductions in the weights of the immune organs due to HS, including the thymus, bursa, and spleen [91,92,93]. As a consequence of HS, Tang and Chen [94] detected significant reductions in B and T lymphocytes, which led to decreases in the production of antibodies, changes in cytokine secretion, and lower numbers of macrophages with a reduced phagocytic ability [95,96]. In 2014, Xu et al.’s study of the chicken spleen detected an increase in the mRNA levels of *Tumor Necrosis Factor alpha* (TNF-α) and *Interleukin-4* (*IL-4*), whereas the levels of *Interferon gamma* (*IFN-γ*) and *Interleukin-2* (*IL-2*) were lowered [97].

Simultaneously, HS triggers heat shock factors and proteins aimed in cellular protection [98]. It is known that fast-responding genes (i.e., pro-inflammatory genes) undergo an inhibition regulated by feedback mechanisms. For instance, *Interleukin-6* (*IL-6*), the most pyrogenic cytokine, was not increased by acute exposition to heat. This is because *Heat Shock Factor 1* (*HSF1*) down-regulates the *IL-6* gene expression [99]. Han et al., in 2010, demonstrated that the response to acute HS in Ross broiler chickens is mediated by a higher activity of *IL-2* in lymphocytes of the spleen [100]. Another study shows how acute exposure to HS enriched the mRNA expression of *IL-4* in the spleen of HS chickens [101].

Quinteiro-Filho et al. highlighted that, in broilers exposed to chronic HS, the plasma levels of IgA and IgG were reduced, consistent with the lower robustness rate of broiler breeds regarding thermal stress [102]. Finally, *Interleukin-17* (*IL-17*) expressed the highest up-regulatory responses to acute HS. Indeed, *IL-17* plays an immunological role in mucosal response and is an activator of the TLR4-dependent means of clearance of pathogenic bacteria [103]. *IL-17* also contributes to the development of adaptive immunity to inflammatory agents [104].

## 8. The Heat Shock Proteins (HSPs) Family

HSPs are characterized by their ability to be induced by HS and, in molecular terms, by the presence of a functional heat shock element in their promoter [105]. HSPs (Figure 3) act as chaperones that bind other proteins, after HS damage, with the purpose of preserving their structure, managing the proteins’ migration across membranes or organelles, or keeping the receptor availability or specific enzyme functionality under control [106].

The expression of the HSP could be physiological or induced by different inputs such as HS. HSPs might be classified in several manners; here, we report the most used classification based on molecular weights: *HSP100*, *HSP90*, *HSP70*, *HSP60*, small HSPs, and chaperonins [107]. Aside from the molecular chaperone activity, small HSPs are also involved in other cellular pathways such as stress tolerance, protein folding, cytoskeletal integrity, and cell cycle [108]. In most cases, HSPs are found in the cytoplasm, but it is also possible to find their presence in the extracellular matrix, in which they could serve as stress alarms and stimulate immune cells [107]. Table 1 briefly summarizes the most studied HSPs and HSFs, the biological processes in which they are involved, and their molecular function in chickens.

## 9. HSP in Commercial Chickens

It is universally known that several stress stimuli, and HS in particular, can influence HSPs’ expression in various chicken tissues. For instance, the quantity of *HSP70* and ubiquitin transcripts increases when testis cells are exposed to high temperatures [109]. In the same way, a relevant increment in the *HSP70* expression in female broiler chicken brains after four days of heat treatment was noticed [110]. Thermal stress caused the induction of *HSP90α* and *HSP90ß* in chicken heart, liver, and spleen, but the *HSP90α* and *HSP90ß* mRNA levels were stable in the brain [111]. The gain expression of HSP in the heart may allow protection in harsh environments. For instance, in the heart tissue of HS broiler *HP60*, *HSP70*, and *HSP90* proteins and their relative mRNAs there is a gain after 2 h of HS, but this diminishes quickly with chronic HS [112]. Indeed, the great diversity in heat shock response among different tissues and different broiler lines is widely known. One study reports that in breast muscle, acute and chronic HS enhanced protein oxidation, but the HSP gene expression remained at physiological levels and the only trend increases were observed in the gene expression of *HSP70* and *HSP90* after acute HS [113]. Thermal stress up-regulates the expression of the *HSP70* gene in two fast-growing broiler strains. It is interesting to note that the expression of the liver *HSP70* gene in heat-stressed Ross broilers was significantly higher than that reported in Cobb [114].

Moreover, Tang et al. reported that two groups of broilers, both under heat stress and one treated with aspirin, showed different responses. The aspirin-treated group showed a greater response to HS in the kidney, with less damage to the tissue, due to the ability of aspirin to induce high expression levels of *HSP47* and *HSP60* [115].

Liu et al. [92] highlighted that HS stimulus is able to raise the expression of *HSP27* (small HSP), *HSP70*, and *HSP90* mRNA in the bursa of Fabricius and the spleen of broilers; these data corroborate with the data on stressed broiler spleen from Slawinska et al.: *HSP25*, *HSP70*, and *HSP90* were up-regulated [101]. Quite the opposite happened in the same heat condition in the thymus: *HSP27* and *HSP90* mRNA were significantly reduced.

In addition, an *HSP70* up-regulation was observed in the serum of heated layers, compared to layers under normal temperature. In some layers, HS substantially altered the gut microbiome [116].

## 10. HSP in Local Chicken Breeds

The genomic tools already discussed have also been applied in several studies in order to verify the HS resistance of local chickens, compared to commercial lines.

The aim of this paragraph is to clarify the pathways that allow local chickens to have a greater resistance ability to HS. Indeed, Cedraz et al. reports the results of a real-time PCR that show the expression levels of several HSPs in the muscle of two local Brazilian chickens (Peloco and Caneluda breeds) and a broiler (Cobb 500) [117]. Thus, under HS the *HSP70* and *HSP90* genes were highly expressed in all genetic groups compared to the control conditions. At the same time, in HS conditions the *HSP70* and *HSP90* genes were significantly more expressed in both local chickens than in the broiler [117].

A microarray-based study highlighted the higher expression of *HSP70*, *HSP90AA1*, and *HSP25* in the testes of a Taiwan egg-type chicken in an HS environment [118]. Moreover, the same authors compared the obtained results with the data from Taiwan meat-type chickens submitted to the same experimental design. The results of the study showed substantial changes in the expression of 28 genes in both types of chickens, and most of these genes were HSP genes. However, the testicular responses to acute HS differed between the two types of chickens: for instance, the *HSP60* gene was highly expressed in HS meat-type chickens; on the contrary, the *BAG3* gene was more expressed in egg-type chickens [119]. *BAG3* and *HSP60* genes both negatively regulate cellular apoptosis; hence, it is very interesting to see how two different chicken breeds have many genes in common, but differ for others when they have to face the same stressed situation.

In a recent study by Sharma et al., the HSPß1 mRNA expression (a member of small HSP family that helps in maintaining the homeostasis of proteins by stabilizing different types of non-native proteins) was evaluated in Punjab Red and Rhode Island Red layers under HS. The results demonstrated a direct association between HS, HSPß1 mRNA expression, and serum HSPß1 concentration [120].

In 2020, Radwan described the possibility of using gene expression data to explain why some traits underwent genetic selection. In fact, the study shed light on improvements in some traits in a native Egyptian chicken line (Fayoumi) exposed to thermal stress, such as heat resistance, by selecting for the desired traits. In the study, Fayoumi chickens were reared either in a normal or a heated environment [121]. Thirty-five-week-old females from the HS group with the best egg production and the strongest eggshells were mated with their male siblings. The same mating program was used with F1 birds to obtain a second generation [119]. Radwan also evaluated the uterus mRNA level of *HSP90* and *ovocleidin 17* (*OC-17*), which were demonstrated to be involved in eggshell strength [121,122]. The results from the real-time PCR show increased levels of *HSP90* and *OC-17* in the two successive generations, meaning that both eggshell strength and heat tolerance were improved thanks to selection in birds raised under conditions of HS [121].

It is noteworthy that some studies used the statistical association between SNPs and thermo-tolerance; indeed, in Zhang et al. the SNP site (C.1388 A>G) of the chicken *Heat Shock Factor 3* (*HSF3*) gene was associated with heat resistance in two chicken lines [123]. Chen et al. showed that the SNP of C.141 G>A in the *HSP90β* gene in chicken had an impact on the HS resistance traits, and the GG genotype represents the most suitable one under HS. The *HSP90β* mRNA expression was shown to be tissue-dependent by qRT-PCR. The expression of *HSP90β* mRNA in the heart, liver, brain, and spleen of Lingshan chickens was significantly superior compared to that of White Recessive Rock, underlining the fact that the Lingshan breed is more adapted to tropical temperatures [124].

A different study in White Recessive Rock chickens was carried out by Kong et al. and highlighted that the C.744C>G SNP in the 5’-flanking region of the *Glucose-Regulated Protein 78* gene (*GRP78*) was significantly correlated with heat tolerance parameters [125]. The gene *GRP78* belongs to the *HSP70* family, is a fundamental chaperon in various animals, works against apoptosis, and allows proteins and organelles to preserve their physiological functionality [126]. Finally, in Kong’s study, a qRT-PCR assay indicated a gain in the *GRP78* mRNA expression in all tissues, which then decreased with chronic HS and peaked at 3 h after HS. In conclusion, the *GRP78* mRNA expression depends on time and tissue [125].

In the same way, Irivboje et al. evaluated genetic polymorphisms in intron 7 and exon 8 of the *HSP90AA1* gene in two exotic Nigerian chicken strains (Brown dominant and Fowl Hyline brown). Several SNPs (A7T, A160T, T223A, and C134T) were detected; however, the SNP A7T, qualified for the association analysis after the Hardy-Weinberg equilibrium test, was not significantly linked to heat resistance features [127].

An important study by Fleming et al. compared several chickens from different breeds and climates of Africa and Northern Europe for a selection signature evaluation; the aim of this work was to understand the adaptation mechanisms of the animals to their local environments. The African chickens were distinguished for their stronger resistance toward stress signaling and angiogenesis, while the Northern European chickens showed more genetic selection toward processes related to energy homeostasis. In Chromosomes 2 and 27, the populations were the most divergent in terms of selection pressure. Moreover, chromosome 27 was involved in heat tolerance in African chickens, while novel insights into unique genomic regions on chromosome 2 could be related to development and environment for Northern European chickens. In conclusion, the study shows in an excellent way how two populations subjected to divergent environments present in their genome the signs of different selection affecting different metabolic pathways [128].

A study in support of the central role of HSPs in facing HS is the recent research of Srikanth et al. Using RNA-Seq methods (in liver and heart tissues), two indigenous chicken ecotypes from Kenya were compared; the animals were sampled from the tropical climate in Mombasa (lowland) and the colder Naivasha (highland) regions in order to investigate the effects of acute (5 h, 35 °C) and chronic (3 days of 35 °C for 8 h/day) HS. Only four different differentially expressed genes (DEGs) were found in all four experimental groups and were identified as HSP70 family member 8 (HSPA8) both in acute and chronic HS and small HSP family member 7 (HSPB7) in acute HS [129]. Moreover, *Fatty Acid–Binding Protein 4* (*FABP4*) was found differentially expressed in chronic HS [129] and also in the hypothalamus of a broiler chicken [130]. The *FABP4* gene is involved in the *Peroxisome Proliferator-Activated Receptor* (*PPAR*) signaling pathway, which is required for energy metabolism and regulating the oxidative stress-induced inflammatory response [129,131]. The enrichment of the *PPAR* pathway indicates that during adverse conditions, such as HS, *PPAR* promotes adaptation events [132].

An interesting study by Wells et al. reports a genome-wide SNP scan using featherless chickens; these chickens carry a single recessive mutation that causes a lack of almost all body feathers, as well as foot scales and spurs, due to a failure of skin patterning during embryogenesis [133,134]. The trait is potentially useful in tropical agriculture due to the ability of featherless chickens to better tolerate heat [135]. Through a cost-effective and labor-efficient SNP array mapping approach, Wells and his groups showed that a nonsense mutation in *FGF20*, related to a loss of a highly conserved region, was completely associated with the featherless phenotype. This aspect was confirmed by in situ hybridization and quantitative RT-PCR assays that revealed a high *FGF20* expression during the early stages of feather placode patterning. Hence, the loss of genetic function highlights the role of *FGF* ligand signaling in feather development and suggests *FGF20* as a novel central player in the development of vertebrate skin; thus, this gene could be considered in crossbreeding in order to obtain featherless chicken lines that are less susceptible to high temperatures [133].

Recently, Te Pas et al. studied the ability of local chickens to rapidly adapt to a hotter environment. Indeed, they analyzed a transcriptome from different tissues (heart, breast muscle, and spleen) in Ethiopian lowland chicken (adapted to lowland thermal conditions) and highland chickens (non-adapted to lowland environments) in relation to the changes in temperature in lowland conditions during the day in the morning, noon, and evening [136]. The highland chickens responded rapidly to HS, and muscle tissue had more HSP regulation in highland chickens, suggesting that muscle tissue is particularly vulnerable to HS more than the other types [136]. Interestingly, through a statistical analysis of RNA-Seq data, evidence of epigenetic mechanisms was found. Indeed, epigenomic regulations on chromatin re-modeling through both DNA methylation and histone (de)acetylation mechanisms were reported. These biological mechanisms are fast regulators, much more so than mutation and selection [137].

## 11. Epigenetic Mechanisms in Chicken HS Response

Epigenetic mechanisms are one of the main options used by the body to better cope with adaptive developmental reprogramming [138]. In fact, the following definition clarifies how epigenetics work: “an epigenetic trait is a stably heritable phenotype resulting from changes in a chromosome without alterations in the DNA sequence.” The above-proposed description of epigenetics can involve the phenotype heritability through either mitosis or meiosis [139].

Epigenetic adaptation assumes that environmental factors—e.g., environmental temperature—can affect the physiological control system during the development phase, which might make modifications in the thermoregulatory process [140]. Genetic investigations in animals revealed that epigenetic markers may be passed down across generations, leaving a mark on the offspring phenotype [141]. Methylation is one of the most considerable epigenetic modification processes, occurring at the DNA level and playing an upstream role in the control of gene expression [142]. A selective pressure of particular stimuli can flow into the modification of the methylation level in the organism. It is known that the methylation level and target can lead to the regulation of the expression of tissue-specific genes [143].

Vinoth et al. showed the DNA methylation patterns of the HSP promoter region in poultry brain tissue in order to assess the number of methylated genes managing thermal adaptation. The authors used two chicken lines embryos (Naked Neck and Punjab Broiler-2) subjected to HS and normal temperature (named Control (C) and thermal-conditioned (TC) embryonic groups, respectively. Chickens developed from these TC and C embryos have been successively subjected to HS and normal temperature, establishing four groups: control normal (CN), control heat exposure (CHE), thermal conditioned normal (TCN), and thermal conditioned heat exposure (TCHE). In the brains of 17-day-old embryos, the HSP gene expression was up-regulated in the TC group of both Naked Neck and Punjab Broiler-2, except for *HSP60*; this implies that the embryonic HS has been stressful for both the chicken breeds. Moreover, the HSP mRNA expression in the brain tissue was lower in TCHE chicks, as a proof of the valuable role played by embryonic heat treatment. CHE chicks exhibited the highest values of all HSP genes. Comparing the expression of HSP between the CHE and TCHE groups, the authors reported an overt heat-stressed state of the CHE chicken. These differences in expression can be related to the different methylation levels in the promoter of the same HSP genes. Indeed, a higher methylation level of *HSP90 α* and *β* and *HSP70* in TCHE, compared to CHE, was found [144]. These results clearly demonstrated how high levels of methylation in the promoter serves as a down-regulator of gene expression. This is in accordance with the study of Gan et al., where in chicken muscle subjected to HS an opposite association between the expression and promoter methylation of *HSP70* was reported [145].

Another study, consistent with the findings of Vinoth et al., described a reduction in *HSP* (*70* and *27*) mRNA expression in different tissues (heart, liver, muscle, spleen, and bursa) in chickens exposed to HS during the late incubation period of embryos [146]. These results corroborate the thesis that a post-hatch heat stress exposure can increase the heat adaptation of chicks.

## 12. A Step over HSPs

In the last five years, different approaches to give a more exhaustive picture of HS in chickens have been established. The subsequent studies have used different statistical analyses and different biological starting points and have obtained varied results, but have maintained the main conductive threads, with the tissue under study being the liver. The liver has become the very first organ target because it represents the metabolic center of the organism. An interesting approach has been used by Jastrebski et al. (2017), who merged the RNA-Seq gene expression approach with the study of metabolome in chronic stressed chickens [147]. In fact, one of the main functions of the liver is to maintain the homeostasis of lipids, sugars, and amino acids in the body. Generally, metabolome and transcriptome data have shown a strong response in the liver. Indeed, many pathways and metabolites had been found to be activated in order to maintain homeostasis and react to issues caused by oxidative stress. In more detail, glycogenolysis and gluconeogenesis (as also shown by Kumar et al. [148]) as well as fat deposition, glycosylation, and glutathione production were increased. Finally, the transcriptome data have shown a trend to slow down the cell cycle in order to allow time for repairing DNA damage caused by HS [147] (Jastrebski et al., 2017).

A change in liver biology has also been noticed by other studies. For instance, Li et al. in 2011 found four new genes involved in the response to HS: *PM20*/*PM21*, *ASB2*, *USP45*, and *TFG*. In particular, the last one was involved in the activation and enrichment of two main molecular pathways, mitogen-associated protein kinase (MAPK) and nuclear factor kappa-light-chain-enhancer of activated B cells (NFKB), in the response of broilers to HS [149]. The results were also corroborated by Bertocchi et al. [150] and especially Coble et al. through RNA-Seq, also showing novel biological processes that were functionally enhanced in HS response: cellular signaling, the endocrine system, and molecular transport. Particularly, the expression patterns of the *CCK*, *TRPC5*, *DIO2*, and *DIO3* genes display a direct link between feed intake, deiodinase activity, and temperature regulation in response to heat [151].

Walugembe et al. performed a GWAS in different ecotypes of chickens from Brazil, Egypt, and Sri Lanka that have to face HS in their natural environment. In particular, Sri Lanka has hot humid climatic conditions that, besides being favorable for the pathological infection of livestock, also present challenging conditions such as HS. Through selection signal methods, the authors found two regions under selection on chromosome 4 across the Sri Lanka ecotypes. This chromosomic region codes for *Toll like receptor 3* (*TLR3*) and *Nuclear factor kappa B subunit 1* (*NFKB1*). *TLR3* usually activates the MAPK and NFKB pathways [152].

Additionally, Sun et al., in 2015, studied the phenomenon of HS in chickens, but used RNA-Seq to identify heat stress-responsive genes in the chicken male white leghorn hepatocellular cell line. Multiple biological processes were affected by the responsive genes, including translation, transcription, chromatin modification, DNA repair, and DNA synthesis. In addition, two signaling pathways were modulated by HS: TGFß-enriched and WNT down-regulated [153].

## 13. Future Perspectives and Conclusions

Due to the enormous improvement of genetic tools, data generation is becoming affordable and effortless, resulting in huge amount of information. In the last 20 years, data should have elucidated the biological system that underpins animal production, health, and welfare traits. This has shed light on the mechanisms and detection of (potential) biomarkers and improved animal breeding strategies.

Several researches have demonstrated that RNA-Seq is the most suitable and promising technology that can give a definitive view of the pathways involved and the role of every single gene in the thermoregulation process. Only after the entire process is elucidated can strong biomarkers be detected. Then, the allelic variants codifying for favorable biomarkers could also be selected in specialized breeds by crossbreeding schemes, including indigenous breeds, or by biotechnological approaches (i.e., gene editing).

With the threat of climate change on animal production, favorable alleles for HS reduction should be introduced in breeding plans. The climatic problem affects both intensive and extensive productive systems; the latter should be more focused on local chicken breeds, which are able to produce in harsh environments. In this context of change, a fundamental role can belong to local breeds—those not selected for commercial production, but that have genetically evolved over time thanks to the selective pressure of the environment in which they have lived. Thus, local breeds could represent an important genetic reservoir of phenotypes more adapted to HS. In this way, according to an increased demand for “ethical food,” biodiversity should have not only an ethical value but economic relevance too.

Finally, this review has summarized the research regarding the use of molecular tools to investigate the HSP response against HS in chickens. The entire process of response to HS is complex and not all molecular steps are completely understood. At this stage, studies working with specific genes give only a partial view of what could happen in the tissues of chickens during HS. In particular, the role played by the different gene families in the HSP-mediated response is not yet clear. Further studies should be carried out to better clarify the mechanisms involved in HS tolerance and to understand if the HSP family (and which HSP family) could be considered a useful biomarker for detecting HS.

## Figures and Tables

**Figure 1 animals-11-00046-f001:**
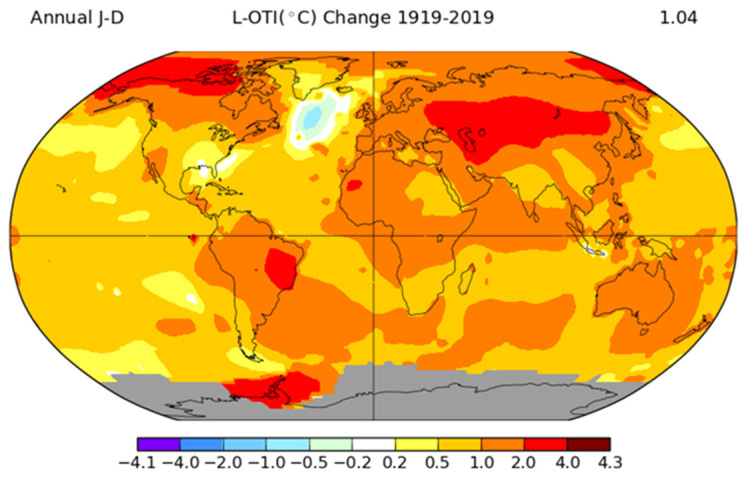
Temperature change (°C) in 2019 compared with the year 1919. Note: grey areas signify missing data (source: figure taken from https://data.giss.nasa.gov/gistemp/ and adapted for illustrative purpose only).

**Figure 2 animals-11-00046-f002:**
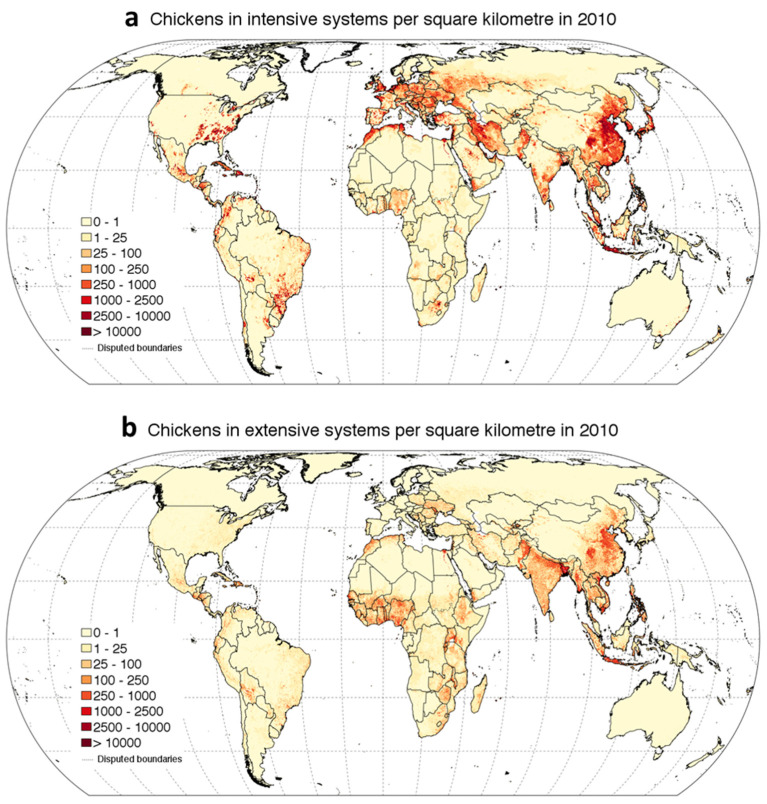
These figures represent the global distribution of chicken in 2010 in intensive (**a**) and extensive (**b**) systems (source: figure taken from https://dataverse.harvard.edu/dataset.xhtml?persistentId = doi:10.7910/DVN/A7GQXG and adapted for illustrative purpose only).

**Figure 3 animals-11-00046-f003:**
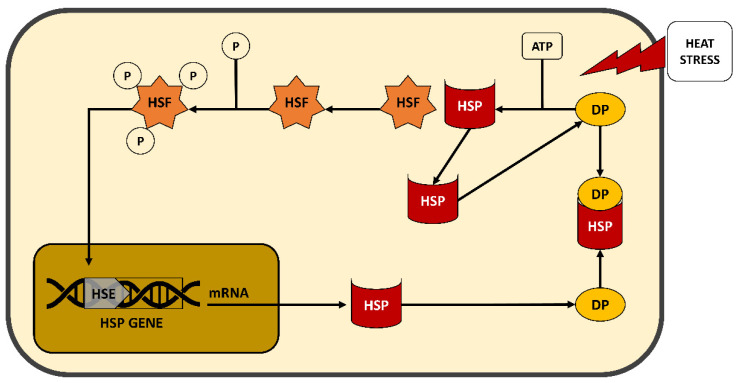
Schematic mechanism of heat shock protein (HSP) in cellular response against heat stress (HS). HSF = Heat-Shock Factor; HSE = Heat-Shock Element; DP = Denatured Protein; P = Phosphate.

**Table 1 animals-11-00046-t001:** Using the Panther Classification System, the gene ontology of the most important genes related to heat stress was performed on two domains (molecular function and biological process). The current red jungle fowl (*Gallus gallus*) genome assembly, GRCg6a.

Gene Name	Mapped Ids	Family Name	Molecular Function	Biological Process
*HSP70* *HSPA2*	NC_006092.5 (GRCg6a)	Heat shock 70 kDa protein 2	ATP binding; heat shock protein binding; unfolded protein binding; ATPase activity;	Vesicle-mediated transport; Chaperone-mediated protein folding; Cellular response to unfolded protein
*HSP90B1*	NC_006088.5 (GRCg6a)	Heat shock protein 90 beta family member 1	Unfolded protein binding	Protein folding
*HSF1*	NC_006089.5 (GRCg6a)	Heat shock factor protein 1	DNA-binding transcription factor activity; RNA polymerase II proximal promoter; sequence-specific DNA binding	Cellular response to heat; Regulation of transcription from RNA polymerase II promoter in response to stress; Transcription by RNA polymerase II; Positive regulation of transcription by RNA polymerase II
*HSF3*	NC_006091.5 (GRCg6a)	Heat shock factor protein 3	DNA-binding transcription factor activity; RNA polymerase II proximal promoter sequence-specific DNA binding	Cellular response to heat; Regulation of transcription from RNA polymerase II promoter in response to stress; Transcription by RNA polymerase II; Positive regulation of transcription by RNA polymerase II
*HSPD1* *HSP60*	NC_006094.5 (GRCg6a)	Heat shock protein family D (Hsp60) member 1	Unfolded protein binding	Protein folding
*SERPINH1* *HSP47*	NC_006088.5 (GRCg6a)	Serpin family H member 1	Endopeptidase inhibitor activity Serine-type endopeptidase activity Protease binding	Proteolysis; Cellular protein metabolic process; Negative regulation of endopeptidase activity

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
