# Peer review of "Emerging Genetic Tools to Investigate Molecular Pathways Related to Heat Stress in Chickens: A Review"

_animals, 2020, doi:10.3390/ani11010046_

Round 1
Reviewer 1 Report
Heat stress is a key issue of general concern in modern broiler production, especially in the context of global climate changes. With the development of next-generation sequencing technology, some emerging genetic tools have been used in the study of heat stress traits in chickens. This manuscript chooses this question as the point, which is relatively new and has certain reference value for future research. However, the emerging genetic tools and heat stress in this manuscript, seem to be divided into two separate points, rather than a whole. This manuscript introduces more principles and limitation of genetic tools and the phenotype of HS, rather than the application of genetic tools in HS research of chicken. Even though the author mentions related content in Chapters 11 and 12, these reviews are limited. Proteomics and metabolomics are also used to study heat stress in chickens, unfortunately these information has not been reviewed in this manuscript. In addition, in the citations of some paper, the authors directly cited review papers instead of original experimental paper, this is not permitting. Therefore, this manuscript is not suitable for publication on animals until these issues are resolved.
Author Response
REVIEWER 1
Comments and Suggestions for Authors
Heat stress is a key issue of general concern in modern broiler production, especially in the context of global climate changes. With the development of next-generation sequencing technology, some emerging genetic tools have been used in the study of heat stress traits in chickens. This manuscript chooses this question as the point, which is relatively new and has certain reference value for future research. However, the emerging genetic tools and heat stress in this manuscript seem to be divided into two separate points, rather than a whole. This manuscript introduces more principles and limitation of genetic tools and the phenotype of HS, rather than the application of genetic tools in HS research of chicken. Even though the author mentions related content in Chapters 11 and 12, these reviews are limited.
AU :Thank you for each of your suggestions, we did our best to improve the manuscript in the direction you suggested.
The aim of this manuscript is to introduce molecular tools (advantages and disadvantages), trying to give an all-round view, avoiding technicalities. Then, we shortly introduce the phenotype that are fundamental for a large part of studies of HS, and to give the idea that the phenotype already used for HS evaluation in chicken are limited. After this, we focused on the application of NGS tools in order to evaluate and understand what happen in genes and their expression when HS occurred. Moreover, we divided this part in 2 chapters (11 and 12) focusing on the importance of local chickens in HS, and the knowledge that new generation tools gave for establishing new biomarker for HS evaluation.
Proteomics and metabolomics are also used to study heat stress in chickens, unfortunately these information has not been reviewed in this manuscript.
AU : According with your suggestions, we expand the last part of the manuscript adding a new chapter. Here, we also try to give importance to metabolomics studies but only in those few studies where metabolic traits were related with genomics data. However, the review is based on new tools to investigated HS in chickens, hence on genomics and phenotype with a particular attention on local chickens; proteomics and metabolomics are important but are macro topics different from those we decided to deal with in this manuscript.
In addition, in the citations of some paper, the authors directly cited review papers instead of original experimental paper, this is not permitting. Therefore, this manuscript is not suitable for publication on animals until these issues are resolved.
AU : Finally, we revised the points in manuscript where we cited directly the review papers. We cited them because the content was extracted from the review content not from a paper cited in that reviews. For this reason, we cited directly the review, that is allowed (i.e. review that we cited, cited other reviews too). Moreover, we cited only 7 reviews and 146 original papers.

Reviewer 2 Report
Emerging genetic tools to investigate molecular pathways related to heat stress in chickens: a review.
Francesco Perini, Filippo Cendron, Giacomo Rovelli, Cesare Castellini, Martino Cassandro, Emiliano Lasagna
Overview and general recommendation:
The heat stress is one of the environmental factors, which decreases the productive performance of poultry and meat quality. Even more, not only in poultry but in general in animal production aspect extended on the while livestock. The current knowledge on this subject indicates that for heat stress are responsible heat shock factor and heat shock proteins involved in the complete step by step sequence of mechanisms and pathways. The knowledge was based on many studies, directly and indirectly, involved in this matter, since QTL mapping (STRs), next GWAS (SNP microarray) and tasks based on next-generation sequencing (omics tools) including WGS, RNASeq and other like epigenetics. The aim of the present review is to summarize papers concerning the identification of HS genetic markers in chickens, using new molecular tools.
The whole paper looks seem to be extensive; there is no typical for original papers part scheme and is more dedicated to the review article. After the short introduction, the paper turns into an exciting part of climate changes. However, the economic factor in chicken production should be described here. How looks like looses in chicken production (eggs and meat) due to the heat stress, even in "normal" breeding on farms extended on climate changes. The next pars were the highly expanded, and I don't know if it's okay, on RNASeq studies. On this basis, it is difficult to do a full verification of the issue. The parts based on the native and commercial chicken breed looks clear and could be linked to the classical discussion. Also noteworthy is the part of epi studies.
Due to the revive of papers and genes indicate in this papes, it could be useful to see a similar analysis of what did by the Mohamad at al., 2020 https://www.mdpi.com/2079-7737/9/4/62 . The authors have done Genes and Pathways for interest genes (heat stress genes and other genes no directly involved in the stress processes) for the review article.
Therefore, I recommend that a minor revision is warranted. I explain my concerns in more detail below. I ask that the authors explicitly address each of my comments in their response.
Additional comments:
- The part called "8. How to establish HS in chicken by molecular tools?" looks highly trivial and brings nothing new to the classic scheme of the analysis.
- The improvement in English is needed. You could find a lot of mistyping in the plain text.
- Is it only a few studies based on QTL mapping? The review should be extended on reports quoted on Animal QTLdb/ Chicken QTLdb - https://www.animalgenome.org/cgi-bin/QTLdb/GG/index.
- The same part comment is dedicated to the genome-wide association studies. This part should be divided on papers based on SNP chips, and WGS (exon?) reports due to different variation as global.
- The authors (in part about phenotype) should be paid attention to studies which try to estimate the heritability of heat stress due to different approaches. It looks like highly adequate to sentence (162) "First, the identification of alleles linked to a specific traits or phenotype required a huge sample sizes (100 to 1, 000) for adequate statistical power of GWAS". Moreover, according to the broad knowledge this senesce could be misleading (lower h2 need the higler population and so on…).
- Did you also verified your study based on papers:
Bertocchi, & Zampiga, Marco & Luise, Diana & Vitali, Marika & Sirri, & Slawinska, Anna & Tavaniello, & Palumbo, & Archetti, Ivonne & Maiorano, & Bosi, & Trevisi, P.. (2019). In ovo Injection of a Galacto-Oligosaccharide Prebiotic in Broiler Chickens Submitted to Heat-Stress: Impact on Transcriptomic Profile and Plasma Immune Parameters. Animals. 9. 1067. 10.3390/ani9121067.
Tavaniello, Siria & Slawinska, Anna & Prioriello, D & Petrecca, V & Bertocchi, M & Zampiga, Marco & Salvatori, Giancarlo & Maiorano, Giuseppe. (2019). Effect of galactooligosaccharides delivered in ovo on meat quality traits of broiler chickens exposed to heat stress. Poultry Science. 99. 10.3382/ps/pez556.
Slawinska, Anna & Zampiga, Marco & Sirri, Federico & Meluzzi, Adele & Bertocchi, M & Tavaniello, Siria & Maiorano, Giuseppe. (2019). Impact of galactooligosaccharides delivered in ovo on mitigating negative effects of heat stress on performance and welfare of broilers. Poultry Science. 99. 10.3382/ps/pez512.
Slawinska, Anna & Mendes, S. & DunisÅ‚awska, Aleksandra & Siwek, Maria & Zampiga, Marco & Sirri, Federico & Meluzzi, Adele & Tavaniello, Siria & Maiorano, Giuseppe. (2019). Avian model to mitigate gut-derived immune response and oxidative stress during heat. Biosystems. 178. 10.1016/j.biosystems.2019.01.007. – wrong date of the paper
Author Response
Please find in the attachment the answers
